# Effect of Resin Infiltration on Enamel: A Systematic Review and Meta-Analysis

**DOI:** 10.3390/jfb12030048

**Published:** 2021-08-16

**Authors:** Madalena Soveral, Vanessa Machado, João Botelho, José João Mendes, Cristina Manso

**Affiliations:** 1Clinical Research Unit (CRU), Centro de Investigação Interdisciplinar Egas Moniz (CiiEM), Egas Moniz—Cooperativa de Ensino Superior, CRL, 2829-511 Almada, Portugal; madalenasvsoveral@gmail.com (M.S.); jbotelho@egasmoniz.edu.pt (J.B.); jmendes@egasmoniz.edu.pt (J.J.M.); mansocristina@gmail.com (C.M.); 2Evidence-Based Hub, Clinical Research Unit, Centro de Investigação Interdisciplinar Egas Moniz (CiiEM), Egas Moniz Cooperativa de Ensino Superior, CRL, 2829-511 Almada, Portugal

**Keywords:** resin infiltration, demineralization, white spot lesions, surface roughness, microhardness, shear bond strength, penetration depth

## Abstract

Subsurface enamel demineralization beneath an intact surface layer or white spots lesions (WSL) can and should be treated with non-invasive procedures to impede the development of a cavitated lesion. We aim to analyze if infiltrative resin improves enamel roughness, microhardness, shear bond strength, and penetration depth. MEDLINE [via Pubmed], Cochrane Central Register of Controlled Trials, Embase, Web of Science, Scholar, and LILACS were searched until May 2021. Methodological quality was assessed using the Joanna Briggs Institute Clinical Appraisal Checklist for Experimental Studies. Pairwise ratio of means (ROM) meta-analyses were carried out to compare the enamel properties after treatment with infiltrative resin on sound enamel and WSLs. From a total of 1604 articles, 48 studies were included. Enamel surface roughness decreased 35% in sound enamel (95%CI: 0.49–0.85, I^2^ = 98.2%) and 54% in WSLs (95%CI: 0.29–0.74, I^2^ = 98.5%). Microhardness reduced 24% in sound enamel (95%CI: 0.73–0.80, I^2^ = 99.1%) and increased by 68% in WSLs (95%CI: 1.51; 1.86, I^2^ = 99.8%). Shear bond strength reduced of 25% in sound enamel (95%CI: 0.60; 0.95, I^2^ = 96.9%) and increased by 89% in WSLs (95%CI: 1.28–2.79, I^2^ = 99.8%). Penetration depth was 65.39% of the WSLs (95%CI: 56.11–74.66, I^2^ = 100%). Infiltrative resins effectively promote evident changes in enamel properties in sound and WSLs. Future studies with long-term follow-ups are necessary to corroborate these results from experimental studies.

## 1. Introduction

Dental caries is an oral condition estimated to affect 2.4 billion people worldwide in 2010 [1,2], while the frequency of white spot lesions (WSLs) varies between 2% and 97% [3,4,5,6]. WSLs were firstly described in 1908 by Black [7]. In the last decades, the prevalence of WSLs has increased as a side effect to fixed orthodontic appliances [8,9,10,11]. Consequently, multiple approaches have been proposed to prevent, manage and treat dental caries [12,13,14], while non-invasive therapies have emerged to treat early signs of WSLs [15,16,17].

Caries infiltration is a minimally invasive technique for the management of smooth surface and proximal non-cavitated caries lesions. Several remineralization products have been presented to this end, such as fluoride, casein phosphopeptide, amorphous calcium phosphate, and microabrasion [7,8]. Low-viscosity light-cured resins are another popular approach [9]. The infiltration of resins creates a diffusion barrier inside the enamel lesion body [18], retarding enamel dissolution [10,11], and the retention loss is unlikely to occur [19].

Clinical evidence points to the partial or total ability of infiltrative resins to mask enamel whitish discoloration [12], despite its clinical efficacy still warrants long-term confirmation [7,13,14,15,16,17]. A previous systematic review of in vitro studies has shown an increase of surface microhardness of WSLs after resin infiltration, and an opposite result in sound enamel [20]. Notwithstanding, there is still uncertainty regarding its efficacy on other characteristics (namely, surface roughness, shear bond strength, and penetration depth). Hence, appraising the available evidence on these characteristics in a systematic manner becomes clinically relevant to understand the potential of this minimally invasive procedure.

Considering the recent increased number of studies, here we present a systematic review assessing the effect of infiltrative resins on surface roughness, microhardness, shear bond strength and penetration depth in permanent teeth with and without enamel lesions.

## 2. Materials and Methods

### 2.1. Protocol and Registration

We registered and approved this systematic review protocol a priori at the National Institute for Health Research PROSPERO database, International Prospective Register of Systematic Review (Available online: www.crd.york.ac.uk, ID number: CRD42019140860) and we reported the review information according to the PRISMA guidelines [18] (Appendix A).

### 2.2. Focused Question and Eligibility Criteria

To answer our main question, we developed a protocol with two PICO questions:“Do infiltrative resin in sound enamel and WSLs improve the surface roughness, microhardness and shear bond strength?” and“What is the penetration depth capacity of the infiltrative resin in WSLs?”

Each question had the following statements: Teeth with sound enamel, and teeth with WSLs or teeth classified with ICDAS 1 or 2 (Population, P); Resin infiltration (Intervention, I); Initial condition or no treatment (Comparison, C); Disappearance or improvement of the surface roughness, microhardness, and shear bond strength (Outcome, O).Teeth with WSLs or teeth classified with ICDAS 1 or 2 (Population, P); Resin infiltration (Intervention, I); Not applicable (Comparison, C); Penetration depth (Outcome, O).

In vitro studies that assess the enamel surface roughness, microhardness, shear bond strength and penetration depth before and after resin infiltration were eligible. In vivo studies were excluded because the methods used for clinical evaluation of those four characteristics mentioned in patients were substantially different from those used on in vitro studies, and there were innumerable variables that we are unable to control, such as the quality of patients’ saliva, their cooperation, and the variation of the techniques and analyses performed. Although the teeth are considered healthy, we assume that they have changed their crystalline structure once they have been subjected to home care products, such as fluoridated toothpaste and oral rinsing solutions or mouthwash being subjected to remineralization with the incorporation of fluoride ions.

Regarding color, no assessment was made because Borges et al. in 2017 [21] systematically evaluated this characteristic in patients. Furthermore, editorial, letters, reviews, thesis, case reports, and case series were excluded. Most studies used profilometers (in Ra) to quantify the surface roughness, and therefore studies that quantified enamel surface roughness using other appliances were not included because it does not allow comparison.

### 2.3. Search Strategy

Seven electronic databases (MEDLINE [via Pubmed], Cochrane Central Register of Controlled Trials, Embase, Web of Science, Scholar, and LILACS) were searched systematically until May 2021. The following search strategies were adjusted to each database: (“infiltrative resin” OR “resin infiltration”) AND (“white spot lesions” OR “white spots” OR “WSL” OR “Enamel demineralization”). In addition, we search manually in Journal of Dentistry, The Journal of Prosthetic Dentistry, Clinical Implant Dentistry and Related Research, Operative Dentistry, Community Dentistry, and Oral Epidemiology, Journal of Conservative Dentistry, and International Journal of Dentistry. The Grey literature was searched using the latter strategy in OpenGray. Any limitation of the publication period and language was applied. Authors were contacted, when necessary, for additional data clarification.

### 2.4. Study Process

Two independent researchers (M.S. and V.M.) screened the title and/or abstract of retrieved studies. Any disagreements were resolved by discussion with a third author (C.M.). The final selection of studies was independently performed by two authors (M.S. and V.M.) who reviewed the selected papers’ full text based on the inclusion criteria mentioned above. For measurement reproducibility purposes, inter-examiner reliability following full-text assessment was calculated via kappa statistics.

A predefined table was used to extract necessary data from each eligible study, including the citation, publication status and year of publication, study design, inclusion/exclusion criteria, number of specimens per group, demineralization process, resin infiltration protocol, surface roughness, microhardness, shear bond strength and penetration depth measurement method. Concerning additional data clarifications, we attempted to contact the corresponding authors twice, with an interval time of 1 week.

### 2.5. Methodological Quality Assessment

Two researchers (M.S. and V.M.) independently assessed the methodological quality of the included studies, following the Joanna Briggs Institute Clinical Appraisal Checklist for Experimental Studies. This assessment tool was adapted from previously published systematic reviews [22,23,24]. The items on the checklist were as follows: (1) clearly mention aim, justification of sample size; (2) sample randomization; (3) blind treatment allocation; (4) possibility of comparison between control and treatment groups; (5) baseline equivalence of control and treatment groups; (6) clearly describe the preparation protocol; (7) clearly report the experimental protocol; (8) measurement method, and adequate statistical analysis. Each item was scored using a 2-point scale: 0—not reported or reported inadequately; and 1—reported and adequate. Any disagreements between the examiners were resolved through discussion with a third author (C.M.).

### 2.6. Statistical Analysis

For continuous data, mean values and standard deviations (SD) were collected to predefined tables prepared to determine the quantity of data. If median and interquartile range were reported in the selected studies, mean and SD were calculated following Hozo’s formula [25]. The random-effect meta-analysis and forest plots were calculated in R version 3.4.1 (R Studio Team 2018) using ‘meta’ package [26], through DerSimonian-Laird random-effects meta-analysis. Firstly, we started by conducting an a priori sensitivity analysis comparing Standardized Mean Difference (SMD) versus Ratio of Means (RoM) meta-analyses. If there are similar results in terms of heterogeneity and significance, RoM was applied as it would allow easier and direct interpretation of the results (reported as percentage) [26]. To investigate sources of heterogeneity, meta-regression analysis was conducted for method, pH, and demineralization time. I^2^ index and Cochrane’s Q statistic were used to assess statistical heterogeneity (*p* < 0.1) and χ2 test calculated overall homogeneity [26]. Substantial heterogeneity was considered when I^2^ statistics exceeded 50% [27]. All tests were two-tailed with alpha set at 0.05 except for the homogeneity test whose significance level cutoff was 0.10 due to the low power of the χ2 test with a limited amount of studies. Overall estimates were reported with 95% confidence interval (CI). For meta-analysis including 10 or more studies, we analyzed publication bias [28].

## 3. Results

### 3.1. Study Selection

The initial database search strategy retrieved 1604 possibly relevant articles. After exclusion of all duplicates, 175 articles were assessed for full paper review eligibility. Among these, 127 articles were excluded with the respective reasons for exclusion detailed in Appendix A. A total of 48 articles fulfilled the inclusion criteria and were selected for further quantitative and qualitative analyses (Figure 1). Good inter-examiner agreement was obtained during full-text screening and article final selection (Cohen’s Kappa: 0.92; 95% CI: 0.89; 0.94).

### 3.2. Characteristics of the Studies

In this systematic review, twenty-three articles reported information about microhardness [27,28,29,30,31,32,33,34,35,36,37,38,39,40,41,42,43,44,45,46,47,48,49], ten evaluated surface roughness, [27,31,37,47,49,50,51,52,53,54], sixteen assessed the penetration depth [40,46,50,55,56,57,58,59,60,61,62,63,64,65,66,67] and eight explored shear bond strength after resin infiltration [32,52,68,69,70,71,72,73].

Overall, twelve studies included bovine teeth (*n* per group = 803) while twenty-five studies included human teeth (*n* per group = 865). Nine studies reported data without specimen demineralization previously to resin infiltration [47,54,55,57,59,61,62,63,64,74] and thirty-nine articles included the demineralization process before the infiltrative resin [21,27,28,29,30,31,32,33,34,35,36,37,38,39,40,41,42,43,44,48,49,50,51,52,53,56,58,60,65,66,67,69,70,71,72,73]. The demineralization process consists of emerging the teeth in a demineralization solution, with pH between 4 and 5 for a period of time that ranges from 1 min to 1200 h, to simulate the formation of WSLs (Table 1).

Nowadays there is only one commercial kit available, ICON^®^ (DMG, Hamburg, Germany), that aims to infiltrate proximal and vestibular lesions [75]. Taking this into account, all studies used the ICON^®^ protocol to do the infiltration of resin.

**Table 1 jfb-12-00048-t001:** Overview of the Included Studies.

Study	Funding	*n*	Specimen Origin	Exclusion CRITERIA	WSLs Preparation (pH for hours)	Outcome Reported
Pancu et al. 2011 (Romania) [42]	NR	10	Human (bicusps or molars)	NR	pH: 4.4 for 120 h	Microhardness (Vicker hardness- special device for microhardness testing with a squared diamond head)
Meyer-Lueckel et al. 2011 (Germany) [59]	DFG: PA 1508/1-1. HML and SP and royalties from DMG, Hamburg	20	Human (molars and premolars)	Active non-cavitated proximal WSL (ICDAS code 2)	Without demineralization	Penetration Depth (Confocal laser scanning microscopy (CLSM)
Paris et al. 2011 B (Germany) [62]	Institute for Immunology, UK-SH, Christian-Albrechts Universitat zu Kiel for providing the CLSM. The Charité—Universitatmedizin Berlin holds US and European patents	19	Human (molars and premolars)	Active non-cavitated proximal lesions scored as ICDAS 2	Without demineralization	Penetration Depth (confocal laser scanning microscope)
Paris et al. 2011 A (Germeny) [64]	DFG: PA 1508/1-2, as part partially by DMG.	16	Human (molars)	Cavitated lesions	Without demineralization	Penetration Depth (Confocal laser scanning microscopy CLSM)
Taher et al. 2012 (Saudi Arabia) [47]	No	10	Human (premolars)	Cracks, restorations, or developmental lesions	Without demineralization	Roughness; Microhardness (microscope with 200 magnification and application of applying a load of 300 g; profilometer)
Torres et al. 2012 (Brazil) [48]	NR	15	Bovine (incisors)	Damaged or not intact enamel	pH: 5 for 16 h	Microhardness (microhardness tester fitted with a 50-g load)
Attin et al. 2012 (Switzerland) [32]	Dentaurum, 3M ESPE, and DMG	12	Bovine (incisors)	NR	pH: NR for 504 h	Shear Bond Strength (universal testing machine)
Veli et al. 2014 (Turkey) [72]	No	20	Human (premolars)	Caries, hypoplastic areas, restorations, and surface abnormalities	pH: 4.8 for 504 h	Shear Bond Strength (universal testing machine)
Ekizer et al. 2012 (Turkey) [70]	No	20	Human (premolars)	Hypoplastic spots, cracks, or gross irregularities	pH: 4.3 for 6 h	Shear Bond Strength (universal testing machine)
Paris et al. 2013 (Germany) [43]	DFG: PA1508/1-2. HML and SP and royalties from DMG, Hamburg.	12	Bovine (incisors)	NR	pH: 4.95 for 1200 h	Microhardness (Vickers hardness with a force (F) of 0.981 N for 10 s)
Paris et al. 2013 (Germany) [61]	DFG: PA 1508/1-1	15	Human (molars and premolars)	Cavitated caries	Without demineralization	Penetration Depth (confocal laser scanning)
Mohammed et al. 2014 (Iraq) [76]	NR	56	Human (premolars)	NR	pH: 4.5 for 120 h	Roughness (profilometer)
Paris et al. 2014 (Germany) [63]	DFG: PA 1508/1-3	9	Human (molars and premolars)	ICDAS codes 0, 1, 2	Without demineralization	Penetration Depth (dual fluorescence confocal microscopy)
Lausch et al. 2014 (Germany) [57]	The Charité Universitätsmedizin Berlin and DMG	17	Human (molars and premolars)	Without active or cavitated WSL	Without demineralization	Penetration Depth (confocal laser scanning)
Gelani et al. 2014 (USA) [56]	No	42	Bovine (incisors)	WSP, cracks, or any other defect	pH:5 for 24 h	Penetration Depth (Confocal Laser Scanning Microscopy and Transverse Microradiography)
Dilber et al. 2014 (Turkey) [69]	NR	15	Human (mandibular lateral teeth)	Hypoplastic areas, cracks, or gross irregularities in enamel	ph:4.3 for 6 h	Shear Bond Strength (Universal testing machine)
Montasser et al. 2015 (Egypt) [41]	No	10	Human (NR)	NR	pH: 4.4 for 504 h	Microhardness (Vickers diamond indenter load of 200 g)
Arslan et al. 2015 (Turkey) [31]	NR	15	Human (central incisors)	NR	pH: 4.5 for 6 h	Roughness; Microhardness (profilometer; Vickers hardness tester with 2 N load)
Min et al. 2015 (South Korea) [60]	Basic Science Research Program through the National Research Foundation of Korea (2013R1A1A2062505)	20	Bovine (permanent anterior teeth)	NR	pH:4.8 for 960 h	Penetration Depth (Optical coherence tomography Confocal laser scanning microscopy))
Vianna et al. 2015 (Brazil) [73]	No	15	Bovine (incisors)	NR	pH:5 for 56 h	Shear Bond Strength (universal testing machine)
Gurdogan et al. 2016 (Turkey) [38]	No	20	Bovine (incisors)	NR	pH: 4 for 2 h	Microhardness (Vickers Hardness tester with 100 gr force)
Abdel-Hakim et al. 2016 (Egypt) [28]	NR	6	Human (molars)	Caries, hypocalcifications, or restorations	pH: 4.4 for 480 h	Microhardness (Vickers michrohardness testing with 200 gm load)
El-zankalouny et al. 2016 (Egypt) [46]	No	7	Human (premolars)	Cracks, caries, or restorations	pH: 4.4 for 96 h	Microhardness; Penetration Depth (Vickers tester with f 50 g; stereomicroscope)
Abdellatif et al. 2016 (Egypt) [29]	NR	11	Human (anterior teeth)	NR	pH: 4.8 for 720 h	Microhardness (Vicker’s microhardness test with load of 200 g)
Baka et al. 2016 (Turkey) [52]	NR	20	Human (premolars)	Hypoplastic areas, cracks, restorations, or gross irregularities	pH: 4.8 for 504 h	Roughness; Shear Bond Strengths (profilometer; a universal testing machine)
Neto et al. 2016 (Brazil) [30]	CAPES, Funcap, and CNPq (Brazilian agencies). Project PON 254/Ric	10	Human (molars)	NR	pH: 4.9 for 16 h	Microhardness (Knoop microhardness)
Horuztepe et al. 2017 (Turkey) [39]	No	45	Bovine (incisors)	Cracks or other surface defects	pH: 4.95 for 672 h	Microhardness (microindentation hardness tester with a 50-g load)
Mandava et al. 2017 (India) [40]	No	20	Human (maxillary central incisors)	Presence of cracks and defects	pH: 4.4 for 96 h	Microhardness; Penetration Depth (Vicker’s microhardness tester with a 300 g load; confocal laser fluorescence microscope)
Aziznezhad et al. 2017 (Iran) [34]	Babol University grant	10	Human (premolars)	Not intact and time of extraction more than 3 months	pH: 4.5 for 6 h	Microhardness (Vickers device with 500 g load)
Prajapati et al. 2017 (India) [44]	No	10	Human (premolars)	Teeth with hypoplasia or incipient carious lesions/WSL	pH:4.4 for 504 h	Microhardness (Vickers microhardness tester with 100 g load)
Sava-Rosianu et al. 2017 (Romania) [65]	Project for young researchers—Programme II-C3-TC-2015	60	Human (premolar)	NR	NR	Penetration Depth (Confocal Laser Scanning Microscopy)
Attia et al. 2018 (Egypt) [77]	NR	20	Bovine (NR)	Cracks or defects in the surface	pH:5 for 24 h	Microhardness (micro-indentation hardness tester (with a 50-g load)
Nabil et al. 2018 (Egypt) [27]	NR	15	Human (anterior teeth)	Cracks and any developmental defects	pH: NR for 1 h	Roughness; Microhardness (profilometer; Vickers Tester with load of 200 g)
Enan et al. 2018 (Egypt) [37]	NR	10	Human (bicuspid)	Cracks and defects	pH: 4.95 for 160 h	Roughness; Microhardness (profilometer; universal testing machine)
Khalid et al. 2018 (Indonesia) [54]	University of Indonesia	10	Human (premolars)	Enamel surface that was attached orthodontic appliance; WSL, defects on the buccal side of enamel; restorations	Without demineralization	Roughness (profilometer)
Yazkan et al. 2018 (Turkey) [49]	Suleyman Demirel University Scientific Research Projects Foundation (2969-D-11)	16	Bovine (incisors)	Caries, fracture, or other defects	pH: 5 for 240 h	Roughness; Microhardness (profilometer; Vickers indenter, with load of 200 g)
Askar et al. 2018 (Germany) [55]	Deutsche Forschungsgemeinschaft (DFG; PA 1508/1-3), and DMG	15	Human (NR)	Active proximal lesions with ICDAS-2, 3 and 5	Without demineralization	Penetration Depth (confocal microscopy)
Aswani et al. 2019 (India) [51]	No	10	Human (anterior teeth)	NR	pH:4.4 for 144 h	Roughness (profilometer)
Enan et al. 2019 (Egypt) [53]	No	30	Human (premolars)	NR	pH:4.95 for 160 h	Roughness (profilometer)
Arora et al. 2019 (India) [50]	No	30	Bovine (premolars)	Caries	pH: 4.5 for 96 h	Roughness; Penetration Depth (optical profilometer)
Theodory et al. 2019 (USA) [66]	Student Government for Graduate and Professional Students at the University of Iowa	15	Human (molars)	NR	pH: 4.3 for 2160 h	Penetration Depth (Confocal Laser Scanning Microscopy)
López et al. 2019 (Brazil) [59]	NR	8	Human (NR)	Cavity lesions, white stains, cracks, or structural alterations and restorations	pH: 5 for 0.5 h	Penetration Depth (Confocal Laser Scanning Microscopy)
Gulec et al. 2019 (Turkey) [71]	NR	20	Human (premolars)	Caries, attrition, fracture, restoration, congenital or surface anomalies, or surface	pH: 4.5 for 22 h	Shear Bond Strength (universal testing machine)
Borges et al. 2019 (Brazil) [68]	FAPESP(2010/16878-7, 2010/17757-9)	30	Bovine (incisors)	NR	pH:5 for 16 h	Shear Bond Strength (Scanning electron microscopy (SEM))
Ayad et al. 2020 (Egypt) [33]	NR	7	Bovine (anteriors)	NR	pH: 4.4 for 96 h	Microhardness (Vickers indenter, with a static load of 200 g)
Behrouzi P et al. 2020 (Iran) [35]	No	15	Human (maxillary central incisors)	Cracks, caries, or mineralization defects	pH: 4.5 for 96 h	Microhardness (Vickers hardness tester with 50 kg load)
El Meligy, 2020 (Saudi Arabia) [36]	No	27	Human (premolars)	ICDAS 1 and 2	pH: 4.5 for 399 h	Microhardness (transversal Vickers hardness with a force of 0.891 N)
Wang et al. 2020 (Brazil) [67]	FAPESP, 2012/13160-3, #2012/18579-2 and 2013/23310-5) CAPES—Brasil	13	Bovine (incisors)	NR	pH: 4.7 for 168 h	Penetration Depth (confocal laser scanning microscopy)

CAPES—Coordenação de Aperfeiçoamento de Pessoal de Nível Superior; CLSM—Confocal Laser Scanning Microscope; CNPq-Conselho Nacional de Desenvolvimento Científico e Tecnológico; DFG—Deutsche Forschungsgemeinschaft; FAPESP—Fundação de Amparo à Pesquisa do Estado de São Paulo; FUNCAP—Fundação Cearense de Apoio ao Desenvolvimento Científico e Tecnológico; H—Hours; HML—Hendrik Meyer-Lueckel; ICDAS—International Caries Detection and Assessment System; NR—Not reported; WSL—white spots lesions; S.P—Sebastian Paris.

### 3.3. Methodological Quality of the Included Studies

Methodological appraisal of the included in-vitro studies using the Joanna Briggs Institute Clinical Appraisal Checklist for Experimental Studies tool is presented in Figure 2 and is detailed in Appendix A. The assessment varied from 7 to 10 (one article with score 7, twenty-two with score 8, twenty-two with score 9 and three with score 10). All included studies showed a clear objective (*n* = 48, 100%) and treated the specimens from the control and experimental group using the same protocol (*n* = 48, 100%). Furthermore, all articles used an appropriate statistical analysis (*n* = 48, 100%) and presented reliable outcomes (*n* = 48, 100%). The majority carefully described the preparation protocol (*n* = 44, 91.7%), and the experimental protocol to characterize the several steps and materials applied (*n* = 39, 81.3%). On the opposite, most articles failed on sample size justification (*n* = 41, 85.4%), in the random assignment of treatment groups (*n* = 15, 31.3%), and only one study reported blindness regarding treatment allocation (*n* = 1, 2.1%).

### 3.4. Clinical Measures

An a priori sensitivity analysis was performed to compare whether both ROM and SMD approaches yielded results in terms of significance and heterogeneity results (Appendix A). Overall, ROM and SMD meta-analyses presented similar significance and heterogeneity degrees, therefore supporting the use of a ROM meta-analytical approach (Appendix A).

#### 3.4.1. Enamel Surface Roughness

Surface roughness before and after resin application was analyzed in both sound enamel and in WSLs, only in human teeth samples (Table 2 and Table 3). It is possible to attest that resin infiltration decreases the 35% (ROM = 0.65, 95% CI: 0.49; 0.85, *p* < 0.0021) the surface roughness in sound enamel and 54% in WSLs (ROM = 0.46, 95% CI: 0.29; 0.74, 0.0012) (Table 2, Appendix A). In both estimates’ heterogeneity was considered high (I^2^ = 98.2% and I^2^ = 98.5%, respectively) (Table 2). Furthermore, multivariate sensitivity analysis demonstrated pH and time of exposure to the demineralizing agent had an important impact on surface roughness (ROM = −1.37, 95% CI: −2.32; −0.42, *p* < 0.005; ROM = 0.01, 95% CI: 0.00; 0.01, *p* < 0.001, respectively) (Table 4).

#### 3.4.2. Enamel Microhardness

Our results showed that resin infiltration significantly reduced by 24%, on average, the microhardness of sound enamel (ROM = 0.76, 95% CI: 0.73; 0.80, *p* < 0.001) and to increase by 68% the microhardness of enamel with WSLs (ROM = 1.68, 95% CI: 1.51; 1.86, *p* < 0.001) (Table 2, Appendix A). In both estimates, the heterogeneity was high (I^2^ = 99.1% and I^2^ = 99.8%, respectively), and there was no publication bias in both analyses (0.8893 and 0.1352, respectively) (Table 2, Appendix A). In particular, the enamel microhardness of sound enamel of human teeth was different compared to bovine teeth (*p* < 0.0188), showing that the enamel microhardness of sound enamel after resin application was lower in human teeth compared to bovine teeth (ROM = 0.58, 95% CI: 0.46; 0.72, and ROM = 0.80, 95% CI: 0.69; 0.92, *p* < 0.0188). Although without any significant difference (*p* = 0.1375), the same pattern was found in human and bovine teeth with WSLs (ROM = 1.59, 95% CI: 1.29; 1.96 and 1.96, 95% CI: 1.64; 2.34, respectively) (Table 3). Still, sensitivity analysis showed no differences based on pH and demineralization time, both using univariate and multivariate analysis (*p* = 0.0188) (Table 4).

#### 3.4.3. Shear Bond Strength

In what concerns shear bond strength, resin infiltration was estimated to reduce 25% the bond strength in sound enamel (ROM = 0.75, 95% CI: 0.60; 0.95, *p* < 0.001) and to increase by 89% the bond strength in WSLs (ROM = 1.89, 95% CI: 1.28; 2.79, *p* < 0.001) (Table 2, Appendix A). In both estimates, heterogeneity was high (I^2^ = 96.9% and I^2^ = 99.8%, respectively) (Table 2). In addition, there was no difference between human and bovine teeth in both analyses (*p* = 0.6221) (Table 3), and only the demineralization time had an important impact on bond strength of teeth with WSLs, both in univariate and multivariate analyses (ROM = 0.00, 95% CI: 0.00; 0.01, *p* = 0.0015 and ROM = 0.00, 95% CI: 0.00; 0.01, *p* = 0.0120, respectively) (Table 4).

#### 3.4.4. Penetration Depth in Caries Lesions

Regarding the penetration depth, only studies with enamel lesions were included. Considering the sound enamel the baseline value as zero, resin infiltration was estimated to penetrate 65.4% of overall lesion (MRAW = 65.4, 95% CI: 56.11; 74.66, *p* = 0.01, I^2^ = 100%) (Table 2, Appendix A). In addition, the longer the application time, the greater the average penetration depth of the resin (Table 3, Appendix A).

## 4. Discussion

### 4.1. Summary of the Main Results

Overall, the present systematic review demonstrates that infiltrative resins effectively change the properties of both sound enamel and WSLs. In sound enamel, infiltrative resins reduced surface roughness, microhardness and shear bond strength. Regarding WSLs, infiltrative resins reduced enamel surface roughness, but increased its microhardness and shear bond strength. Furthermore, estimates point to an average penetration depth capacity of 65% by this type of resin.

### 4.2. Quality of the Evidence and Potential Biases in the Review Process

There are limitations inherent to the included studies. The protocols to create artificial WSLs differ in the pH of the demineralized agent used and etching time, and this could be a source of heterogeneity. Our sensitivity analyses via meta-regression confirmed that pH significantly influences surface roughness in WSLs and resin penetration depth after infiltration technique, and the etching time affected surface roughness and shear bond strength in WSLs. Thus, these protocol variations might affect the interaction with superficial crystals [78] and, therefore, might have contributed to the heterogeneity in the estimates. Additionally, our estimates included both studies on bovine and human teeth. Although this could undermine the consistency of the results, sensitivity analyses only showed an effect on microhardness in sound enamel specimens, and for the remaining analyses there was no significant impact. Nevertheless, future studies shall look for a harmonization of the protocol of WSL creation as well the specimen origin towards a consistent study methodology. In addition, most studies lacked an appropriate rationale for sample size calculation and group allocation of specimens, and these should be accounted for in future studies.

This systematic review also presents some strengths that are worth discussing. A strict protocol was followed with a guideline-based methodology and extensive scientific search. To the best of our knowledge, this is the first systematic review to demonstrate how much infiltrative resins can improve surface roughness, shear bond strength and penetration depth in permanent teeth with and without enamel lesions. Regarding microhardness, although one systematic review had estimated the impact of microhardness on WSLs and sound enamel [20], our approach explored for the first time the effect of pH of the demineralized agent used, etching time, and tooth origin through meta-regression sensitivity analysis on this enamel characteristic. In addition, we included 10 and 16 new studies on sound enamel and WSL, respectively (350% and 229% of the total number of included studies [20], and more than 800 specimens (170% of the total number of specimens) comparing to previous systematic review [20]. Furthermore, by using two meta-analytical approaches and the effort to detect and mitigate potential sources of heterogeneity, we are secure with the effect sizes across the included studies.

### 4.3. Agreements and Disagreements with Other Reviews or Studies and Clinical Relevance

Resin infiltration technique is a minimally invasive therapy to WSLs [78] with a pre-etching phase onto the lesion to improve penetration ability [28,77]. In addition, this penetrative role is enhanced by its methacrylate-based resin matrix containing BisGMA (bisphenol A diglycidil dimethacrylate) and TEGDMA (triethylene glycol dimethacrylate) [43,49], which confer low viscosity to the resin [35].

Analyzing our results on surface roughness, both sound enamel and WSLs decreased the roughness after resin-infiltration application (35% and 54%, respectively), and these results have clinical importance. The oral cavity constantly undergoes a dynamic demineralization-remineralization cycle that promotes natural healing processes [79]. On the one hand, oral biofilm and dietary acids can contribute to create porous lesions on enamel, and, on the other hand, saliva, sealants, antibacterials, fluoride, and a controlled diet with less sugar and starchy foods promote a non-demineralizing environment [80]. Hence, infiltrative resins may play a role not only interventional but also preventive in the enamel roughness resulting throughout life.

Microhardness is a linear enamel characteristic based on the local calcium content [41], and this parameter can be used to assess the increase or reduction of percentage of enamel porosity [28]. Although resin infiltration might increase the microhardness, the establishment of the polymeric chain does not always happen in the entire lesion [81]. Therefore, the inability of a strong intermolecular bond plus the non-infiltration of the resin in the entire enamel lesion can prevent the full recovery of the enamel microhardness. Our results confirmed that the resin infiltration cannot return the microhardness of WSLs to that of sound enamel, although it may restore 68% of it. This result is in agreement with one systematic review that had shown a 3.66 mean difference increase compared with untreated samples [20]. Furthermore, our results fully comply with this previous work with a similar level of heterogeneity. Yet, as above mentioned, our results explored other characteristics that until today had not been in an evidence-based manner.

Shear bond strength concerns the amount of force required to break the adhesive/adherend interface connection [82], and our results showed this characteristic decreases 25% in sound enamel and increases 89% in WSLs. These results are consistent with the literature [52,70]. Firstly, enamel lesions have a degree of porosity with high permeability, allowing the infiltration of the resin. This ultimately results in micromechanical interdigitation strengthening, and therefore increases shear bond strength [52,70]. Secondly, the decrease of shear bond strength on sound enamel may be justified by the low quality of the enamel surface and lack of resin tags for mechanical interlocking [52].

The resin infiltration of WSLs with low viscosity resin results in a hybrid enamel with resin tags that impregnates the interprismatic enamel and reinforces the hard tissue [60,83,84]. Despite its qualities, not the whole portion of the lesion is filled with the resin [40,65,66]. Furthermore, increasing time of application of resin infiltration improves depth penetration [59,63]. The ‘Washburn equation’ describes the time-dependent as an important characteristic to advantage the viscosity, surface tension, and contact angle and allows the resin penetration into porous solids [68]. Comprehensively, our results highlight 65,35% of overall enamel lesions were filled with resin, and the longer the application time, the greater the average penetration depth. Our results are in fully agreement with the literature [43,50,77].

The animal origin of the samples may explain the heterogeneity observed. Bovine teeth are often used in this type of studies, due to their similarities to human teeth [85]. Bovine teeth have a larger crystalline diameter, and their calcium distribution is more homogenous [86]. This species also has a lower fluoride concentration and increased porosity [87]. Nevertheless, the calcium/phosphorus ratio of the mineral removed from the enamel surfaces during demineralization, as well as the remineralization characteristics are similar [88]. Furthermore, caries progression in these two specimens is identical, and the inhibition and composition of biofilm formed are alike [89]. In addition, bovine enamel has approximately the same microhardness as human enamel [80], and no significant differences in bond strength between human and bovine enamel were found [90]. All in all, the reader must bear in mind the aforementioned differences and similarities when analyzing the results of the present review.

## 5. Conclusions

Resin infiltration significantly changes surface roughness, microhardness and shear bond strength in both sound enamel and WSLs. In sound enamel, infiltrative resins decrease 35% of surface roughness, 24% of microhardness and 25% of shear bond strength. In WSLs, enamel surface roughness reduced 54% after infiltrative resins application, but increased 68% and 89% its microhardness and shear bond strength, respectively. Furthermore, estimates point to an average penetration depth capacity of 65% in WSLs. These enamel characteristics can be affected by specimen, pH, and etching time. Future studies with homogeneous methodologies are warranted to confirm these results.

## Figures and Tables

**Figure 1 jfb-12-00048-f001:**
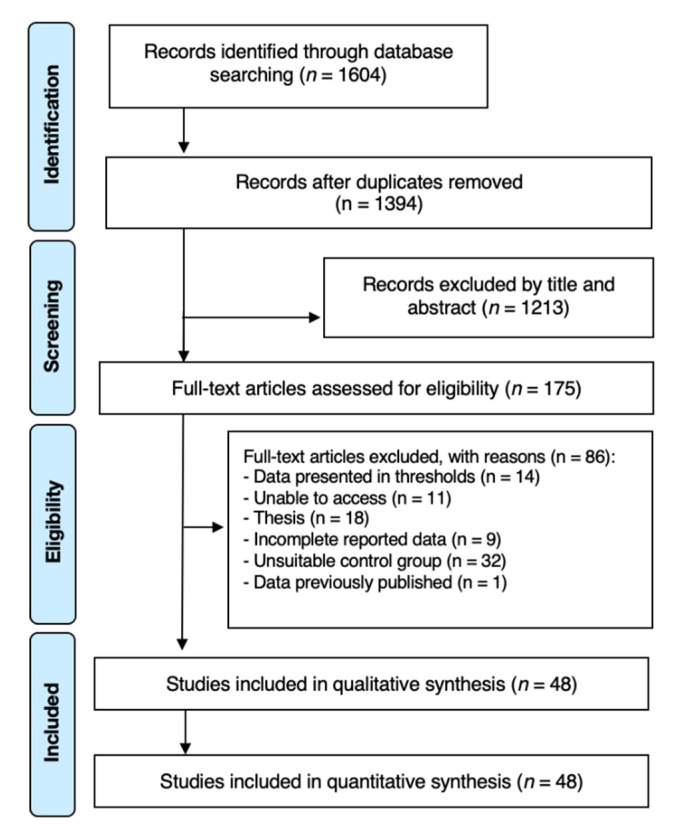
PRISMA flow-chart representing the results of the workflow to identify eligible studies.

**Figure 2 jfb-12-00048-f002:**
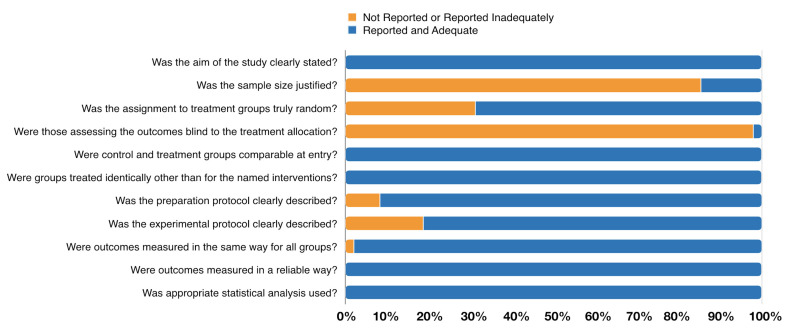
Assessment of the risk of bias in the included studies according to the percentage of the scores attributed to each evaluated study.

**Table 2 jfb-12-00048-t002:** Sound enamel and white spot lesions according to surface roughness, enamel microhardness, bond strength and penetration depth.

Variable	*n* Studies	ROM	95% CI	*p*-Value	I^2^ (%)	Egger Test
**Surface roughness**						
Sound Enamel	5	0.65	0.49; 0.85	0.0021	98.2	-
WSL	8	0.46	0.29; 0.74	0.0012	98.5	-
**Enamel microhardness**						
Sound Enamel	14	0.76	0.73; 0.8	<0.0001	99.1	0.8893
WSL	23	1.68	1.51;1.86	<0.0001	99.8	0.1352
**Shear Bond strength**						
Sound Enamel	6	0.75	0.60; 0.95	<0.0001	96.9	-
WSL	8	1.89	1.28; 2.79	<0.0001	99.8	-
**Penetration depth**						
Sound Enamel	15	65.39	56.11; 74.66	0.01	100.0	0.4712

CI—Confidence Interval; N—number; ROM—Ratio of Means; WSL—White Spot Lesions. Bold-face denotes significance.

**Table 3 jfb-12-00048-t003:** Sensitivity analysis of type of samples using meta-regressions.

Specimen Origin	*n*	B	95% CI	I^2^ (%)	*p*-Value
**Surface Roughness of Sound Enamel**					
Human	4	0.65	0.49; 0.85	98.8	-
Bovine	0	-	-	-	
**Surface Roughness of WSL**					
Human	8	0.46	0.29; 0.74	98.5	-
Bovine	0	-	-	-	
**Enamel Microhardness of Sound Enamel**					
Human	8	0.58	0.46; 0.72	99.0	0.0188
Bovine	6	0.80	0.69; 0.92	99.3	
**Enamel Microhardness of WSL**					
Human	15	1.59	1.29; 1.96	99.5	0.1375
Bovine	8	1.96	1.64; 2.34	99.4	
**Shear Bond Strength of Sound Enamel**					
Human	4	0.75	0.57; 0.99	97.2	0.9958
Bovine	2	0.75	0.41; 1.37	97.7	
**Shear Bond Strength of WSL**					
Human	5	1.74	1.14: 2.65	98.7	0.6221
Bovine	3	2.20	0.93: 5.29	98.8	
**Penetration Depth**					
Human	20	63.65	52.21; 75.09	99.3	0.5589
Bovine	6	71.22	48.57; 93.87	99.8	
ICDAS	16	62.37	46.12; 78.61	99.2	0.4500
WSLs	10	70.29	57.68;82.90	99.7	
Application of infiltrate for less than 1 min	4	49.17	33.36; 64.97	94.1	0.0270
Application of infiltrate for 3 min	18	65.36	55.77; 74.96	99.4	
Application of infiltrate for 5 min	4	81.45	63.88; 99.01	96.5	

CI—Confidence Interval; N—number; ROM—Ratio of Means; WSL—White Spot Lesions.

**Table 4 jfb-12-00048-t004:** Sensitivity analysis of pH and demineralization time using meta-regressions.

Univariate				Multivariate		
Charateristics	B	95% CI	*p*-Value	B	95% CI	*p*-Value
**Surface roughness of WSLs**						
pH	−0.51	−2.96; 1.93	0.6806	−1.37	−2.32; −0.42	0.00451
Demineralization time (hours)	0.00	−0.00; 0.01	1.8929	0.01	0.00; 0.01	<0.0001
**Enamel microhardness of WSLs**						
pH	0.42	−3.42; 0.67	0.0641	0.42	−0.03; 0.00	0.0647
Demineralization time (hours)	0.00	−0.01; 0.00	0.6230	0.00	−0.01; 0.00	0.8729
**Shear Bond Strength of WSLs**						
pH	−0.59	−6.98; 7.62	0.9781	−0.60	−2.02; 0.82	0.4098
Demineralization time (hours)	0.00	0.00; 0.01	0.0015	0.00	0.00; 0.01	0.0120
**Penetration depth of WSLs**						
pH	−6.24	−54.77; 42.29	0.8010	−33.63	−0.88; 290.10	0.0432
Demineralization time (hours)	0.00	−0.02; 0.02	0.9969	0.00	−0.01; 0.01	0.8542
Time of application of resin infiltrate (minutes)	7.68	1.17; 14.18	0.0207	24.12	13.16; 35.07	<0.0001

CI—Confidence Interval.

## Data Availability

Data sharing is not applicable to this article.

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
