# Peer review of "Effect of Resin Infiltration on Enamel: A Systematic Review and Meta-Analysis"

_jfb, 2021, doi:10.3390/jfb12030048_

Round 1

Reviewer 1 Report

In my opinon, after the performed changes the manuscript is now suitable for publication.

Author Response

We appreciate your time reviewing our paper and your words. 

Reviewer 2 Report

Dear Authors, first of all I would like to congratulate You on your work. The topic Is of great clinical relevance. However, I believe that the article could be improved. Please, take a note of some suggestions.

Introduction- this section is too long and should be more focused on the topic in question.

Discussion- this paragraph should be rearranged. It is very chaotic. Please do not repeat information from Introduction and try to be more focused. Rewrite this section using following paragraphs: main results and clinical relevance; comparison with other studies; advantages and disadvantages of the study; conclusions and suggestions for future studies.

Conclusion are very short .

I would suggest to cite this paper to improve your review :

Tooth fragment reattachment technique on a pluri traumatized tooth DOI10.4103/0972-0707.92613

I believe that your manuscript would have much more relevance after suggested improvements.

Author Response

Introduction- this section is too long and should be more focused on the topic in question.

Our answer: We understand and agree with your remark. We improved our introduction section.

Discussion- this paragraph should be rearranged. It is very chaotic. Please do not repeat information from Introduction and try to be more focused. Rewrite this section using following paragraphs: main results and clinical relevance; comparison with other studies; advantages and disadvantages of the study; conclusions and suggestions for future studies.

Our answer: We thank you for mentioning this.  By re-reading the discussion and based on the reviewer's review, we fully agree with the reviewer's view. In fact, the discussion, mainly in the “Agreements and disagreements with other reviews or studies and clinical relevance”, the information was repeated throughout the manuscript. Thus, we focused the discussion on the specific results of this systematic review.

Furthermore, since the discussion has a previously defined thread, we prefer to use the subsections mentioned by the reviewer  but in another order, ie: 4.1. Summary of the main results (rather than “Main results”), 4.2. Quality of the evidence and potential biases in the review process (rather than “advantages and disadvantages of the study”), 4.3. Agreements and disagreements with other reviews or studies and clinical relevance (rather than “Comparison with other studies, and clinical relevance”), and conclusions. The suggestions for future studies were mentioned throughout the discussion section. Nevertheless,  if the editor considers that the subsections should be adapted, we are fully available.

Conclusion are very short .

Our answer: We appreciate this remark. We have added more information by state: “Resin infiltration significantly changes surface roughness, microhardness and shear bond strength in both sound enamel and WSLs. In sound enamel, infiltrative resins decrease 35% of surface roughness, 24% of microhardness and 25% of shear bond strength. In WSLs, enamel surface roughness reduced 54% after infiltrative resins application, but increased 68% and 89% its microhardness and shear bond strength, respectively. Furthermore, estimates point to an average penetration depth capacity of 65% in WSLs. These enamel characteristics can be affected by specimen, pH and etching time. Future studies with homogeneous methodologies are warranted to confirm these results.”

I would suggest to cite this paper to improve your review :

Tooth fragment reattachment technique on a pluri traumatized tooth DOI10.4103/0972-0707.92613

I believe that your manuscript would have much more relevance after suggested improvements.

Our answer: Thank you for this suggestion. After a careful reading, we agree with its relevance and was cited in the introduction section.

Reviewer 3 Report

The paper is well written, and the subject highlighted is one of major interest for minimally invasive dentistry.

The protocol of data gathering and organization is well structured and described and the results were futher extensivelly processed.

However, I suggest the following modifications/additions to the original manuscript:

Line 37: the grammatical form needs to be verified;

Line 64- use „in vitro” instead of „vitro”;

Line 96. Even it is less used, it is still possible to evaluate in vivo some of the analyzed characteristics ( surface roughness, by using a highly precision impression , followed by an epoxidic replica); in this respect, to rephrase the repective statement

Line 101- instead of „oral elixisirs” I suggest „oral rinsing solutions or mouthwash”

In the description of the analyzed study, it would be useful for the reader to find data regarding the methods that have been used in each case for the evaluation of the microhardness, shear bond strength and penetration depth. These ca be added in an additional  column in the table 1.

It is needed to add information also regarding  the materials used for infiltration, cited by each paper (are they commercial materials, experimental materials)? If the authors made pertinent comments regarding the differences in the behaviour of the dental structure- in relation to the origin- it is also needed to consider the infiltrative material in the interaction tooth-infiltrative resin

Author Response

The paper is well written, and the subject highlighted is one of major interest for minimally invasive dentistry.

The protocol of data gathering and organization is well structured and described and the results were further extensively processed.

Our answer: We appreciate your time reviewing our paper and your words.

However, I suggest the following modifications/additions to the original manuscript:

Line 37: the grammatical form needs to be verified;

Line 64- use „in vitro” instead of „vitro”;

Line 96. Even it is less used, it is still possible to evaluate in vivo some of the analyzed characteristics ( surface roughness, by using a highly precision impression , followed by an epoxidic replica); in this respect, to rephrase the repective statement

Line 101- instead of „oral elixisirs” I suggest „oral rinsing solutions or mouthwash”

Our answer:  We have revised all sentences as per your suggestion:

Line 37 now reads: “Demineralization process of the tooth is multifactorial, and although the organic acids (...)”

Line 64 now reads: “A previous systematic review of in vitro studies has shown an increase of surface (...)”

Line 96 now reads: “In vivo studies were excluded because the methods used for clinical evaluation of those four characteristics mentioned in patients were substantial difference of those used in studies in vitro and there were innumerable variables that we are unable to control, such as the quality of patients' saliva, their cooperation, and the variation of the techniques and analyses performed.”

Line 101 now reads: “Although the teeth are considered healthy, we assume that they have changed their crystalline structure once they have been subjected to home care products, such as fluoridated toothpaste and oral rinsing solutions or mouthwash, being subjected to re-mineralization with the incorporation of fluoride ions.”

In the description of the analyzed study, it would be useful for the reader to find data regarding the methods that have been used in each case for the evaluation of the microhardness, shear bond strength and penetration depth. These can be added in an additional column in the table 1.

Our answer: Thank you for the recommendation. The data regarding the evaluation methods in each variable have been added to table 1.

It is needed to add information also regarding  the materials used for infiltration, cited by each paper (are they commercial materials, experimental materials)? If the authors made pertinent comments regarding the differences in the behaviour of the dental structure- in relation to the origin- it is also needed to consider the infiltrative material in the interaction tooth-infiltrative resin

Our answer: We appreciate this remark. Currently, only one commercial kit [Icon® (DMG, Hamburg, Germany) kit] is available to do caries infiltration of both proximal and vestibular lesions. Also, no study used an experimental material. Additional information was added to the methods to clarify this detail.

This manuscript is a resubmission of an earlier submission. The following is a list of the peer review reports and author responses from that submission.

Round 1

Reviewer 1 Report

Dear Authors,

in my opinion this imanuscript is focused on an interesting subject.

Nevertheless, I find several overlapping with the subject reported in recently published reviews, such as:

https://doi.org/10.1016/j.jebdp.2020.101405

In my opinion, the manuscript should be significantly improved before being considered suitable for publication. Here below I report my main suggestions:

- Widening the subject of the study

- Increasing the number of analysed works (records excluded only by title and abstract, see Figure 1, is huge)

- Articles excluded as unrelated (Appendix S2) is very high

- Improving the methodology. For instance, it would be useful to see the results of the independent analysis performed by the 2 researchers reported in Appendix S3, with details about the disagreements and how they were solved with the discussion with a third researcher.

- Improve the comparison and the related discussion on results obtained on bovines and humans. What are the main differences amongst the experiments performed on these two classes? Are they comparable? What are the compositional and anatomical differences and similarities?

I strongly suggest to deeply revise the manuscript before resubmitting it.

Author Response

Dear Special Issue Editors

Prof. Dr. Neamat Hassan Abubakr and Prof. Dr. Karl Kingsley,

We are pleased to resubmit our manuscript entitled “Effect of Resin Infiltration on Enamel: a Systematic Review and Meta-analysis" (Manuscript ID jfb-1251003.R1).

We have considered all commentaries and incorporated changes in the newly revised version of the manuscript. We have taken longer to ensure that every point has been considered.

Please find enclosed a tracked-changes draft of the manuscript and in addition a point-by-point rebuttal to all comments raised as outlined below. We hope that you find our responses satisfactory in addressing the criticism and suggestions.

We hope the revised manuscript will be in an acceptable format for your journal.

Reviewer 1

Dear Authors, in my opinion this manuscript is focused on an interesting subject.

Our answer: We appreciate your time reviewing our paper and your words.

Nevertheless, I find several overlapping with the subject reported in recently published reviews, such as: https://doi.org/10.1016/j.jebdp.2020.101405

Our answer: We appreciate this remark. In fact, the referred systematic review has one overlapping aim, that is, to investigate the effect of resin infiltration on surface hardness of white spot lesions (in our review this is one of our abroad aim that accounts for 4 enamel characteristics). However, there are clear differences from our meta-analytical evaluation to theirs. In our study, 23 articles were included compared to the 7 articles included in the latter systematic review. On the other hand, our results are calculated as Ratio of Means, which allows an easier and more direct interpretation of the results, since they are reported in percentages. All in all, both meta-analyses point to the same direction, that is, resin infiltration significantly reduced healthy enamel microhardness and increased enamel microhardness with white spot lesions. Lastly, the previously published systematic review did not explored, through sensitivity analysis, the effect of the type of tooth used (bovine versus human), publication bias, pH and demineralization time protocols in the analyses. For these reasons, we considered it essential to include microhardness assessment in our systematic review.

We further discussed the differences to this latter study in our Discussion section:

  • “A previous systematic review of vitro studies has shown an increase of surface micro-hardness of WSLs after resin infiltration, an contrary results in sound enamel [11].”. (Lines 60-62)
  • “Only one systematic review had shown similar results to ours regarding microhardness, with 3.66 mean difference increase compared with untreated samples [11]. Nevertheless, this previous review did not explore the impact of the type of tooth used (bovine versus human), and pH and demineralization time protocols to create WSLs. ” (Lines 404-408)

In my opinion, the manuscript should be significantly improved before being considered suitable for publication. Here below I report my main suggestions:

- Widening the subject of the study

Our answer: Although several variables are studied in the literature regarding this topic, most do not present sufficient studies or concise data for a meta-analysis to be carried out. Thus, only variables that allow their statistical quantification were studied to obtain reliable results.

- Increasing the number of analyzed works

Our answer: Although we desired to increase the number of analyzed works, the included ones derived from a strict protocol with a guideline-based methodology and extensive scientific search. For this reason, all excluded and included articles were justified and clearly present in the supplementary file.

- Articles excluded as unrelated.

Our answer: We appreciate this remark. We reviewed the articles excluded as unrelated and we clarified the exclusion reasons.

- Improving the methodology (results of the independent analysis performed by the 2 researchers reported in Appendix S3, with details about the disagreements and how they were solved with the discussion with a third researcher.)

Our answer: We have added the results of inter-examiners reliability as per your instructions. “Good inter-examiner reliability was confirmed at the RoB assessment (kappa score = 0.94, 95% CI: 0.84; 1.00).” (Lines 260-261).

- Improve the comparison and the related discussion on results obtained on bovines and humans. (What are the main differences amongst the experiments performed on these two classes? Are they comparable? What are the compositional and anatomical differences and similarities?)

Our answer: We appreciate pointing out this detail. We have added information regarding the diferences and similarities between bovine and human teeth by stating: “The animal origin of the samples may explain the heterogeneity observed. Bovine teeth are often used in this type of studies, due to its similarities to human teeth [17].  Bovine teeth have larger crystalline diameter, and their calcium distribution is more homogenous [18]. This species also has a lower fluoride concentration and increased porosity [19]. Nevertheless, the calcium/phosphorus ratio of the mineral removed from the enamel surfaces during demineralization, as well as the remineralization characteristics are similar [20]. Furthermore, caries progression in these two specimens is identical, and the inhibition and composition of biofilm formed are alike [21]. Also, bovine enamel has approximately the same microhardness as human enamel [17], and no significant dif-ferences in bond strength between human and bovine enamel were found [22]. All in all, the reader must bear in mind the aforementioned differences and similarities when analysing the results of the present review.” (Lines 426-437)

Reviewer 2 Report

The manuscript titled "Effect of Resin Infiltration on Enamel: a Systematic Review and Meta-analysis" is well written and shows an in depth understanding of the enamel roughness, micro hardness, shear bond  strength and penetration depth by the infiltration of resin. The paper is well structured and systematic search of the literature data has been done to compile the results. Just a few doubts

1.In the literature selected, did they mention which teeth was used for testing? 1The enamel thickness may vary depending on the morphology of the teeth and anomalies.

2. The introduction need to be still elaborate and explain more on the statement of problem.

The authors need to write a strong statement of problem for the research and they need to add up on the hypothesis. For instance, the first paragraph is very short in regards to the background. The authors need to add up few datas on the white spot lesions and their significances. In the middle paragraph it would be better to add few lines mentioning the statement of problem with few valid relevant literature to support the problems. There is very few information on the resins and their types. The authors need to add more literature data on the resins and their effect on the enamel. 

Author Response

Dear Special Issue Editors

Prof. Dr. Neamat Hassan Abubakr and Prof. Dr. Karl Kingsley,

We are pleased to resubmit our manuscript entitled “Effect of Resin Infiltration on Enamel: a Systematic Review and Meta-analysis" (Manuscript ID jfb-1251003.R1).

We have considered all commentaries and incorporated changes in the newly revised version of the manuscript. We have taken longer to ensure that every point has been considered.

Please find enclosed a tracked-changes draft of the manuscript and in addition a point-by-point rebuttal to all comments raised as outlined below. We hope that you find our responses satisfactory in addressing the criticism and suggestions.

We hope the revised manuscript will be in an acceptable format for your journal.

Reviewer 2.
The manuscript titled "Effect of Resin Infiltration on Enamel: a Systematic Review and Meta-analysis" is well written and shows an in depth understanding of the enamel roughness, micro hardness, shear bond  strength and penetration depth by the infiltration of resin. The paper is well structured and systematic search of the literature data has been done to compile the results. Just a few doubts
- In the literature selected, did they mention which teeth was used for testing? 1The enamel thickness may vary depending on the morphology of the teeth and anomalies.
Our answer: We appreciate pointing out this detail. We have added a column clarifying data on the type of tooth used in each study. Most studies report they used sound teeth without detailing structural changes, and teeth abnormalities were an exclusion criterion in all studies. Furthermore, overall enamel samples were prepared with specific dimensions to facilitate testing and analysis. Differences in measurements could be a source of bias and this fact was added into the discussion: The animal origin of the samples may explain the heterogeneity observed. Bovine teeth are often used in this type of studies, due to its similarities to human teeth [17].  Bovine teeth have larger crystalline diameter, and their calcium distribution is more homogenous [18]. This species also has a lower fluoride concentration and increased porosity [19]. Nevertheless, the calcium/phosphorus ratio of the mineral removed from the enamel surfaces during demineralization, as well as the remineralization characteristics are similar [20]. Furthermore, caries progression in these two specimens is identical, and the inhibition and composition of biofilm formed are alike [21]. Also, bovine enamel has approximately the same microhardness as human enamel [17], and no significant dif-ferences in bond strength between human and bovine enamel were found [22]. All in all, the reader must bear in mind the aforementioned differences and similarities when analysing the results of the present review.” (Lines 426-437)

2. The introduction need to be still elaborate and explain more on the statement of problem.
The authors need to write a strong statement of problem for the research and they need to add up on the hypothesis. For instance, the first paragraph is very short in regards to the background. The authors need to add up few datas on the white spot lesions and their significances. In the middle paragraph it would be better to add few lines mentioning the statement of problem with few valid relevant literature to support the problems.
Our answer: We understand and agree with your remark. We have detailed the statement by rephrasing to: “The demineralization process of the tooth is multifactorial, although organic acids pro-duced by pathogenic bacteria, which subsequently leach out of calcium and phosphate ions and leave porosities on the enamel surface, were pointed as the main risk factor [3,4]. White spot lesions (WSLs) were firstly described in 1908 by Black as “occasional white or ashy gray spots that were small and covered with the ordinary glazed surface of the enamel, so that an exploring tine will glide over them the same as over the perfect enamel”[5]. In the last decades, the prevalence of WSLs in enamel has been increased as a side-effect to fixed orthodontic appliances [6–9]. On one hand, appliances challenge oral hygiene [10]. On the other hand, the etching of enamel surfaces with phosphoric acid during the application of orthodontic-fixed appliances causes iatrogenic enamel defects [11]. “ (Lines 32-42)

There is very few information on the resins and their types. The authors need to add more literature data on the resins and their effect on the enamel.
Our answer: We appreciate this suggestion. We added this information by stating: “The concept of caries infiltration was first established as a micro-invasive technique for the management of smooth surface and proximal non-cavitated caries lesions. Therefore, resin infiltration creates a diffusion barrier inside the enamel lesion body rather than only on top as sealants [15]. This process retards enamel dissolution [10,11] and the retention loss is unlikely to occur [16]. “ (Lines 54-58)

Reviewer 3 Report

Dear Authors, first of all I would like to congratulate You on your work. The topic Is of great clinical relevance. However, I believe that the article could be improved. Please, take a note of some suggestions.Introduction- this section is too short, I would suggest these paper to improve this article :

1)Flowable resin and marginal gap on tooth third medial cavity involving enamel and radicular cementum: A SEM evaluation of two restoration techniques

 Statistical analysis is very basic

Taking everything in consideration, I strongly suggest that you rearrange the manuscript (especially Introduction and Discussion) and enlarge the section introduction.   

Author Response

Dear Special Issue Editors

Prof. Dr. Neamat Hassan Abubakr and Prof. Dr. Karl Kingsley,

We are pleased to resubmit our manuscript entitled “Effect of Resin Infiltration on Enamel: a Systematic Review and Meta-analysis" (Manuscript ID jfb-1251003.R1).

We have considered all commentaries and incorporated changes in the newly revised version of the manuscript. We have taken longer to ensure that every point has been considered.

Please find enclosed a tracked-changes draft of the manuscript and in addition a point-by-point rebuttal to all comments raised as outlined below. We hope that you find our responses satisfactory in addressing the criticism and suggestions.

We hope the revised manuscript will be in an acceptable format for your journal.

Reviewer 3.

Dear Authors, first of all I would like to congratulate You on your work. The topic Is of great clinical relevance.

Our answer: We appreciate your time reviewing our paper and your words.

However, I believe that the article could be improved. Please, take a note of some suggestions.

Introduction- this section is too short, I would suggest these paper to improve this article. 1)Flowable resin and marginal gap on tooth third medial cavity involving enamel and radicular cementum: A SEM evaluation of two restoration techniques

Our answer: Regarding the proposed article, and despite it being very interesting, unfortunately does not cover the topic of our systematic review, thus we were not able to cite it in the introduction section.

Statistical analysis is very basic

Our answer: Regarding the statistical analysis, we have followed the Cochrane and PRISMA guidelines to make this part. We tried to add all necessary information in order to be replicable in further studies.

Taking everything in consideration, I strongly suggest that you rearrange the manuscript (especially Introduction and Discussion) and enlarge the section introduction.  

Our answer: We understand and agree with your remark. We improved our introduction section adding the following informations:

“The demineralization process of the tooth is multifactorial, although organic acids produced by pathogenic bacteria, which subsequently leach out of calcium and phosphate ions and leave porosities on the enamel surface, were pointed as the main risk factor [3,4]. White spot lesions (WSLs) were firstly described in 1908 by Black as “occasional white or ashy gray spots that were small and covered with the ordinary glazed surface of the enamel, so that an exploring tine will glide over them the same as over the perfect enamel”[5]. In the last decades, the prevalence of WSLs in enamel has been increased as a side-effect to fixed orthodontic appliances [6–9]. On one hand, appliances challenge the oral hygiene during weeks [10]. On the other hand, the etching of enamel surfaces with phosphoric acid during the application of orthodontic-fixed appliances causes iatrogenic enamel defects [11]. “ (Lines 32-42)

“The concept of caries infiltration was first established as a micro-invasive technique for the management of smooth surface and proximal non-cavitated caries lesions. Therefore, resin infiltration creates a diffusion barrier inside the enamel lesion body rather than only on top as sealants [15]. This process retards enamel dissolution [10,11] and the retention loss is unlikely to occur [16]. “ (Lines 54-58)